# Immune Modulation with Oral DNA/RNA Nanoparticles

**DOI:** 10.3390/pharmaceutics17050609

**Published:** 2025-05-04

**Authors:** Ulpan Kart, Aigul Raimbekova, Sergey Yegorov, Gonzalo Hortelano

**Affiliations:** Department of Biology, School of Sciences and Humanities, Nazarbayev University, 53 Kabanbay Batyr Ave., Astana 010000, Kazakhstan; ulpan.kart@nu.edu.kz (U.K.); aigul.raimbekova@nu.edu.kz (A.R.); sergey.yegorov@nu.edu.kz (S.Y.)

**Keywords:** nanoparticles, DNA, RNA, chitosan, LNP, immune response, drug delivery

## Abstract

The oral delivery of DNA/RNA nanoparticles represents a transformative approach in immunotherapy and vaccine development. These nanoparticles enable targeted immune modulation by delivering genetic material to specific cells in the gut-associated immune system, triggering both mucosal and systemic immune responses. Unlike parenteral administration, the oral route offers a unique immunological environment that supports both tolerance and activation, depending on the formulation design. This review explores the underlying mechanisms of immune modulation by DNA/RNA nanoparticles, their design and delivery strategies, and recent advances in their application. Emphasis is placed on strategies to overcome physiological barriers such as acidic pH, enzymatic degradation, mucus entrapment, and epithelial tight junctions. Special attention is given to the role of gut-associated lymphoid tissue in mediating immune responses and the therapeutic potential of these systems in oral vaccine platforms, food allergies, autoimmune diseases, and chronic inflammation. Despite challenges, recent advances in nanoparticle formulation support the translation of these technologies into clinical applications for both therapeutic immunomodulation and vaccination.

## 1. Introduction

The discovery of the double helix model of DNA by Watson and Crick in 1953 provided a conceptual framework for understanding the mechanisms of DNA replication and genetic inheritance. Although appropriate tools were lacking then, this breakthrough laid the foundation for the future development of gene therapy [1]. Since the first clinical gene therapy trial in 1990, the field has advanced significantly, leading to numerous clinically approved products [2]. The rapid development and global acceptance of mRNA and adenovirus-based vaccines during the COVID-19 pandemic further underscored the potential of nucleic acid-based therapies, propelling them into the mainstream of modern medicine in just over one year [3,4]. Despite ongoing regulatory discussions in the European Union that aim to distinguish RNA technology from existing gene therapy policy, mRNA, like DNA, belongs to the new generation of nucleic acid-based drugs in the arsenal of physicians to treat medical conditions [5].

Gene therapy often relies on viral vectors due to their high efficiency of transgene expression [2]. However, as demonstrated by the adenoviral COVID-19 vaccines, vectors tend to elicit strong immune responses [6]. While this immunogenicity can be advantageous in vaccine development, it presents challenges in treating chronic conditions that require long-term and repeated administrations [7]. Non-viral vectors, such as DNA/RNA nanoparticles (NPs), offer a less immunogenic alternative, though they typically exhibit lower expression efficiency [8,9].

The immune system responds to stimuli by evaluating it within the existing context. For instance, an injected formulation that breaches the skin is often perceived as a danger signal, triggering vascular permeability and a subsequent inflammatory response [10]. In this case, the injected formulation may be identified as part of the threat, thereby provoking an immune reaction in the host. In contrast, oral administration benefits from the unique immunological environment of the gut. The gastrointestinal tract is routinely exposed to dietary antigens and commensal microbes, which allows it to distinguish between harmful and harmless stimuli. This capacity for immune tolerance has been clinically exploited in the treatment of food allergies and inflammatory conditions [11,12]. This function opens the possibility to modulate immune responses to orally administered biologicals. At the same time, the gut must vigilantly monitor and prevent the absorption of harmful pathogens [13]. Therefore, an accurate and balanced regulation of luminal content is critical for maintaining immune and physiological homeostasis [14,15,16].

This review discusses how orally delivered DNA/RNA NPs interact with the immune system, with a focus on biological barriers, delivery strategies, and emerging therapeutic applications.

## 2. Gut Immunology: Target of Oral Nanoparticles

The gastrointestinal (GI) tract plays a central role in immune regulation, serving as both a physical and immunological mucosal barrier [17]. It harbors the gut-associated lymphoid tissue (GALT), the largest immune organ in the body, which maintains a delicate balance between immune tolerance toward non-pathogenic antigens and protective responses against pathogens [18,19]. Structurally, GALT includes Peyer’s patches, isolated lymphoid follicles in the intestinal wall, intraepithelial lymphocytes, and immune cells within the lamina propria [20,21]. Although not anatomically part of GALT, mesenteric lymph nodes filter antigens, and immune cells from the gut via the lymphatic system, initiating adaptive immune responses based on the incoming signals [20].

The immune responses in the gut represent a complex interplay of innate and adaptive immune components, including the mucosal epithelium and tissue-resident innate and adaptive immune cells [22]. In the context of orally delivered nanoparticles, the small intestine is typically regarded as the primary site of uptake due to its extensive surface area, the presence of villi and microvilli, and the activity of specialized antigen-sampling cells [23,24,25]. The more neutral pH of the small intestine, compared to the acidic environment of the stomach, further promotes NP stability, enabling effective uptake and subsequent interaction with the GALT [23,26].

The intestinal epithelium is protected by a mucus layer, containing antimicrobial peptides and immunoglobulin A (IgA) [27]. Beyond serving as a physical barrier against gut microflora, the gut epithelium shapes immune interactions by modulating the presentation of antigens to underlying immune cells [28]. The specialized microfold cells (M cells), localized to Peyer’s patches, and dendritic cells (DCs) that extend processes through the epithelium, actively sample luminal antigens. These antigen-presenting cells deliver antigens to T cells in the mesenteric lymph nodes, initiating either an inflammatory or tolerogenic response depending on the immunological context [29]. Regulatory T cells (Tregs) promote tolerance to commensal microbes and dietary antigens, whereas Th17 cells are critical for mucosal defense [30].

The innate immune cells, such as macrophages, DCs, and innate lymphoid cells (ILCs) detect pathogen-associated molecular patterns (PAMPs) via pattern recognition receptors (PRRs), such as Toll-like receptors (TLRs) and nucleotide-binding oligomerization domain-like receptors (NLRs). These receptors initiate signaling cascades that promote inflammation and facilitate pathogen clearance [31,32].

The gut microbiota also plays a key role in modulating immune responses. It produces metabolites, such as short-chain fatty acids, which facilitate anti-inflammatory signaling pathways and Treg differentiation [33]. Gut dysbiosis can disrupt this homeostasis, contributing to inflammation and immune dysfunction at the mucosal interface [34,35]. Thus, the unique immunological environment of the gut needs to be carefully considered when designing nanoparticle-based strategies for immune modulation.

## 3. Nanoparticle Systems for Oral Nucleic Acid Delivery

NPs are carriers with sizes ranging from 10 to 1000 nm, designed to deliver and protect the cargo from degradation [36]. Due to the physical-chemical characteristics of NPs, such as high-loading capacity, biocompatibility, and biodegradability, they have become widely used across pharmaceutical applications. The capabilities of NPs make them available to carry various therapeutic agents including drugs, nucleic acids, or peptides [37]. DNA/RNA NPs specifically refer to nanosystems in which nucleic acids are used as functional cargo. Their structure is tailored to protect the nucleic acids from degradation and enable targeted delivery to specific cells or tissues [38].

Advancements in material sciences and analytical techniques made it possible to design different nanocarriers with diverse properties. These include liposomes, polymeric NPs, inorganic NPs, and lipid NPs [39]. Liposomes, the most extensively studied nanocarriers for targeted drug delivery, have typically spherical morphology and range from 50 to 500 nm in size. They are composed of natural or synthetic lipid bilayers [40]. Polymeric NPs are are colloidal carriers made from natural polymers (e.g., chitosan, albumin, alginate, cellulose), synthetic polymers (e.g., poly(lactic-co-glycolic acid) (PLGA), polyethylenimine (PEI), polycaprolactone (PCL)), or hybrid polymers [41,42,43]. Inorganic nanoparticles are highly stable and hydrophilic carriers with unique physicochemical properties. Common types include silver, gold, iron oxide, and zinc oxide NPs [44]. Lipid NPs are spherical structures formed from ionizable lipids, which are positively charged at a low pH and neutral at a physiological pH, allowing for pH-responsive delivery [45].

Compared to conventional delivery approaches, NP-based systems offer several advantages for oral administration, including enhanced stability, targeting, and absorption in the gastrointestinal tract [46,47] (Figure 1). However, they also present unique challenges that must be addressed for effective translation. A summary of the main strengths and limitations of various NPs for the oral delivery of nucleic acids is provided in Table 1.

## 4. Biological Barriers to the Oral Delivery of DNA/RNA Nanoparticles

The oral delivery route of DNA/RNA NPs has gained considerable attention as a non-invasive and patient-friendly approach for genetic therapy and vaccination, offering significant potential to enhance treatment accessibility and compliance [92,93]. However, the GI tract presents a challenging environment that can compromise the stability and absorption of these nanoparticles. Factors such as low pH, digestive enzymes, and physical barriers significantly limit their clinical applications (Figure 2) [94,95].

### 4.1. pH Gradient

In healthy adults, the pH of the GI tract ranges from 1.5 and 3.5 in the stomach, increases to about 5 and 6 in the duodenum, and rises further to 7 and 8 in the ileum and colon [96]. These changes impact the structural stability and functionality of DNA/RNA NPs. Acidic conditions in the stomach promote hydrolysis, protonation, and depurination, destabilizing the NPs and reducing their efficiency [97,98].

Moreover, the acidic pH can result in the protonation of the nucleic acids, altering their charge and increasing electrostatic repulsion. This renders the nanoparticles even more unstable and can significantly decrease their efficacy [98]. Once the nanoparticles enter the small intestine, where the pH rises to 5–8, the sudden change in conditions can further stress the particles and affect their structural integrity [26].

To overcome these challenges, different protective strategies have been developed. One of the most effective strategies is encapsulating nucleic acids within lipid NPs. The lipid layer serves as a protective shield against the acidic environment of the stomach [99]. Another approach involves using pH-sensitive polymer coatings. These structures remain intact under acidic conditions but dissolve in the more neutral pH of the intestine, enabling site-specific release [100].

### 4.2. Enzymatic Degradation

Apart from the pH gradient, GI contains a variety of enzymes that contribute to the degradation of orally delivered nucleic acids. Nucleases such as DNase I and RNase, found in saliva, gastric fluids, and pancreatic secretions, actively degrade DNA and RNA [101]. Among these, RNA is especially vulnerable, as RNases efficiently cleave RNA strands, posing a significant obstacle for RNA-based therapies such as siRNA and mRNA [102].

In addition, protective NPs composed of proteins can themselves be degraded by digestive enzymes in the stomach [103]. Recent studies have shown that pepsin can directly degrade nucleic acids, breaking them into smaller fragments [104]. Further, in the small intestine additional enzymes, such as phosphatases and endonucleases, further break down the nucleic acids into mononucleotides, significantly reducing their therapeutic efficacy [105].

These challenges could be addressed in several ways. Chemical modifications, such as 2′-O-methylation and phosphorothioate linkages, demonstrated improvement in the stability of nucleic acids by enhancing their resistance to nuclease activity while maintaining their function [106]. Another practical approach is encapsulation, where nucleic acids are enclosed within lipid or polymeric carriers. These carriers act as a physical barrier, protecting the nucleic acids from enzymatic degradation during their journey through the GI and ensuring their safe delivery to target sites [107].

### 4.3. Mucus Layer

The mucus layer of the GI is composed of highly glycosylated mucins, which are negatively charged and demonstrate viscoelastic properties [108]. These properties make it harder for NPs to diffuse and penetrate. The negatively charged mucins can interact with the phosphate backbones of nucleic acids, effectively “trapping” the NPs and leading to their premature clearance [109].

In addition, the rapid turnover of the mucus layer poses another significant challenge. With a residence time of only 50–270 min, NPs have a limited opportunity to adhere to or penetrate the mucus before it is replaced. This short window significantly reduces their ability to reach the underlying ECs for absorption [110].

Addressing these challenges has focused on modifying the surface of NPs to improve their mobility and retention within the mucus layer. Surface modifications, such as coating NPs with hydrophilic polymers, e.g., polyethylene glycol (PEG), have proven effective in minimizing mucin interactions and increasing NP diffusion through the mucus [111]. For example, PEG-modified exosomes have significantly improved permeability through the mucus layer and increased stability in acidic conditions [33].

Another solution might be to use mucoadhesive and mucus-penetrating coatings. Mucoadhesive polymers increase the retention time of NPs by adhering to the mucus layer, increasing their chances of reaching the ECs [112]. In contrast, mucus-penetrating coatings, such as zwitterionic or hydrophilic polymers, enable NPs to bypass the mucus layer entirely, facilitating efficient delivery to the target cells [113].

### 4.4. Epithelial Tight Junctions

To reach systemic circulation after oral administration, DNA/RNA NPs must cross the intestinal epithelium, a highly selective barrier [114]. This epithelium consists of tightly packed cells held together by tight junctions (TJs), which strictly regulate what can pass between the cells. TJs are especially effective at blocking larger molecules, like most DNA/RNA NPs, which can exceed the 500 Da size threshold [115].

While TJs are a significant challenge, there is another possible pathway for NPs to the M cells in Peyer’s patches. These specialized cells are designed to sample antigens and transport them to the underlying GALT [116]. However, M cells are limited in number, making up less than 1% of the epithelial surface. This limited presence reduces their effectiveness for NP absorption [117].

Several strategies have been developed to overcome these epithelial barriers. One promising method to overcome these barriers is using permeation enhancers. These agents temporarily loosen the TJs, allowing NPs to pass between the cells while preserving the overall integrity of epithelium [118]. Another innovative approach is targeting M cells directly. By modifying NPs with specific ligands or antibodies that bind to M cell receptors, NPs uptake can be increased. This targeted delivery enhances transcytosis and promotes more efficient delivery to immune tissues within the gut [119].

In summary, the oral delivery of DNA/RNA NPs is hindered by a complex combination of physiological and biochemical barriers, including the pH gradient, enzymatic degradation, mucus entrapment, and epithelial TJs. These factors significantly reduce the stability and absorption of nucleic acid therapeutics. Overcoming these challenges requires the careful consideration of NP design, surface modifications, and targeting strategies to enable efficient and safe delivery through the GI tract.

## 5. Interaction of Nanoparticles with Peyer’s Patches in the GALT

Peyer’s patches, primarily localized to the ileum, consist of aggregated lymphoid follicles, and are covered by a specialized follicle-associated epithelium that includes antigen-sampling M cells [120]. M cells lack the thick mucus layer, and microvilli present in other intestinal epithelial cells (ECs), making them more accessible for luminal antigen uptake. By targeting Peyer’s patches, DNA/RNA NPs can induce localized immune modulation within the GALT [121].

M cells are specialized for transcytosis of particulate antigens across the epithelium into the subepithelial dome of Peyer’s patches, where immune cells are concentrated [122]. Once internalized, NPs are taken up by DCs, which process and present their cargo to T cells. This uptake can be enhanced through the surface functionalization of NPs with targeting ligands such as mannose or chitosan, promoting receptor-mediated interactions with DCs [123].

In addition to cellular uptake, DNA/RNA NPs may engage PRRs expressed on epithelial or resident immune cells within Peyer’s patches. For example, DNA NPs can activate TLR9, while RNA NPs are recognized by TLR3, TLR7, or TLR8, contributing to innate immune signaling and potential adjuvant effects [124,125]. These interactions, along with dendritic cell-mediated presentation to T cells and the induction of Tregs, are central to achieving immune modulation and tolerance [126,127,128].

## 6. Design and Applications of Nanoparticles for Inducing Immune Tolerance and Activation

Understanding the immunological pathways triggered by different orally delivered modalities can help design NPs to either activate or down-regulate the immune system. These outcomes can be tailored through various combinations of physicochemical properties, immune cell targets, and the disease or physiological context [129].

Generally, immune activation occurs when NPs deliver antigens or immunostimulatory molecules that elicit inflammatory responses, leading to the recruitment and priming innate and adaptive immune cells [130]. This effect can be further amplified by co-loading NPs with immune adjuvants such as CpG oligodeoxynucleotides (TLR9 agonists) or poly(I:C) (TLR3 agonist), enhancing innate signaling and promoting robust antigen-specific responses. The co-delivery of mucosal adjuvants, such as cholera toxin subunit B, can also facilitate B cell activation and mucosal antibody production [131].

However, an immunomodulatory effect could be elicited using NPs delivering antigens in an anti-inflammatory context, such as by inducing the differentiation of naïve T cells into FoxP3+ Tregs with the help of tolerogenic DCs localized to Peyer’s patches [132,133]. Notably, to enhance tolerogenicity, specific molecular signals have been used to program DCs into tolerogenic states. Examples of such tolerogenicity-inducing cues are vitamin D analogs or cytokine TGF-β, which could be packaged into a nanoparticle alongside the antigens [134]. Nevertheless, another strategy exploiting the tolerogenic nature of commensal intestinal flora is applying metabolites, such as short-chain fatty acids (SCFAs), that promote Treg differentiation (Figure 3) [133].

The design of nanoparticles is central to directing these outcomes. Smaller NPs (<200 nm) are more efficiently taken up by Peyer’s patches, while positively charged particles may improve mucosal adhesion and also risk triggering unwanted inflammatory responses [135]. Biodegradable materials such as chitosan or PLGA can be engineered for slow and sustained antigen release, enabling prolonged immune stimulation or gradual induction of tolerance depending on therapeutic goals [136,137]. Moreover, functionalization with ligands, such as mannose or DEC-205 antibodies, enables targeted delivery to M cells or tolerogenic DCs, promoting selective immune activation or regulation based on the nanoparticle payload context [138].

**Figure 3 pharmaceutics-17-00609-f003:**
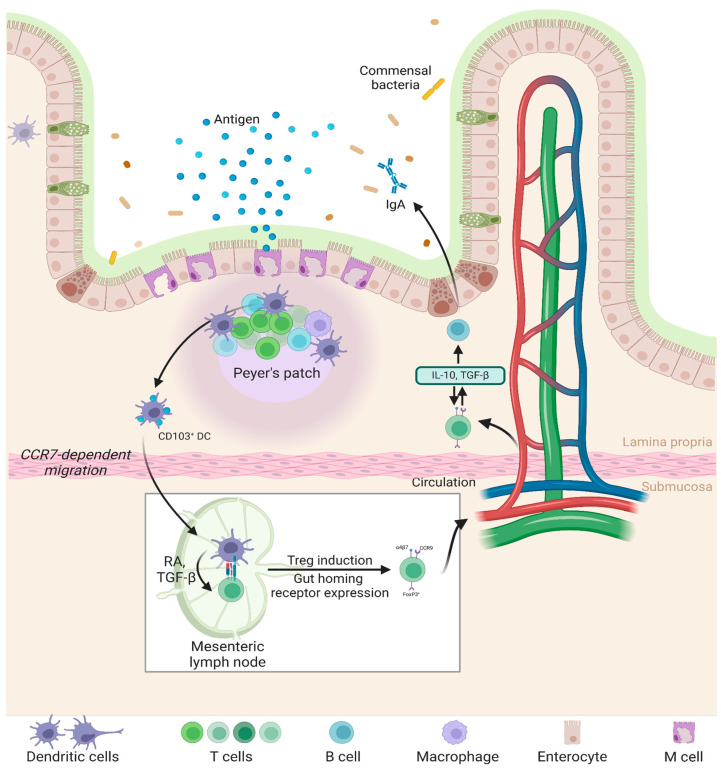
Mechanisms of oral tolerance prevent inflammatory responses to ingested antigens by promoting immune regulation in the gut. Peyer’s patches and mesenteric lymph nodes (mLNs) facilitate antigen uptake, where DCs present them to T cells, inducing Tregs. These Tregs produce TGF-β and IL-10, suppressing inflammation and maintaining gut homeostasis [139,140].

## 7. DNA/RNA Nanoparticles for Oral Vaccination

The oral delivery of NP-based vaccines has become a promising strategy to overcome the challenges caused by the physiological and immunological barriers of the GI tract [21]. NPs are a delivery vehicle protecting antigens from enzymatic degradation and providing targeted delivery to specific sites [141]. Antigens can be encapsulated into NPs or conjugated outside the NPs to promote immune activation [142]. However, in vivo studies have demonstrated that NP-encapsulated antigens in vaccine strategies elicit stronger mucosal immune responses compared to surface-coated antigens [143]. Furthermore, their physicochemical parameters, such as small size, surface charge, and high surface area-to-volume ratio, allow NPs to protect the antigens from intestinal barriers and increase antigen bioavailability [143,144]. These properties make NP-based vaccines effective for inducing systemic and mucosal immunity, positioning them as an attractive solution for oral vaccine delivery [145].

### 7.1. Liposome-Based Vaccines

Liposomes were the first DNA and RNA carrier that received FDA approval, and it is the first delivery platform proposed from proof-of-concept to clinical application [146]. Their amphiphilic structure allows for the encapsulation of hydrophilic molecules like nucleic acids in their aqueous core and lipophilic components in the lipid bilayer, making them highly versatile to formulate [147]. Liposomes protect nucleic acids from enzymatic degradation in the GI and lead to antigen expression [148]. For instance, the oral administration of cationic liposomes DNA vaccine encapsulating *Mycobacterium tuberculosis* antigen 85A (Ag85A) demonstrated antigen-specific mucosal and systemic immunity against tuberculosis [91]. Similarly, Liu et al. reported that cationic liposomes delivering a DNA vaccine encoding the M1 gene of influenza A virus induced both humoral and cellular immune responses in mice and provided protection against respiratory infections [149]. In veterinary applications, liposome-based DNA vaccines have shown efficacy in reducing intestinal colonization of *Salmonella enteritidis* and increasing antibody titers against Newcastle disease in chickens [150].

### 7.2. Lipid NP-Based Vaccines

Lipid NPs have become highly effective carriers for oral drug delivery. They are derived from biocompatible polymers, biodegradable lipids, and oils to improve bioavailability by enhancing permeability or bypassing the first-pass effect [151]. Lipid NPs are promising vehicles due to their low toxicity and have been used for delivering DNA, RNA, and drugs [152]. They enhance drug absorption by preventing precipitation during intestinal dilution, increasing solubilization, reducing enzyme activity, inhibiting efflux transporters, and promoting chylomicron production and lymphatic transport [153].

Lipid NPs are formulated using biodegradable lipids with low toxicity, offering protection against chemical degradation and allowing flexible drug release profiles [154]. Lipid NPs could consist of solid lipids (SLNPs), which are stabilized by surfactants and provide stability but have limitations like poor drug loading and expulsion during storage [155]. Nanostructured lipid particles (NLCs) address these problems by incorporating solid and liquid lipids, creating an imperfect crystalline matrix that enhances drug accommodation [156].

During the COVID-19 pandemic lipid NP were effective in delivering SARS-CoV-2 spike protein. After clinical trials, two lipid NPs mRNA-based vaccines Moderna (mRNA-1273) and Pfizer-BioNTech’s (BNT162b), were given Emergency Use Authorization and EMA conditional approval from the FDA for intramuscular administration [58,157]. The success of mRNA lipid NP vaccines opened new possibilities to develop additional vaccine designs. Mohammadi et al. designed orally vaccinating self-replicating RNA (saRNA) lipid NPs which induced a strong Th1-biased immunity and effectively neutralized SARS-CoV-2 variants B.1.1.7 (alpha) and B.1.617 (delta). Additionally, splenocytes from vaccinated mice showed significant IFN-γ, IL-6, and TNF-α secretion, highlighting the potential of oral lipid NP-based vaccines to boost immune response [158]. During the same period, Keikha et al. developed an oral saRNA lipid NP vaccine, transfected into *Lactobacillus plantarum*. The vaccine successfully induced strong mucosal and systemic immune responses in mice, producing IgG and IgA antibodies capable of neutralizing Alpha and Delta variants. The highest expression of spike protein was observed in the small intestine, confirming effective oral delivery [159].

### 7.3. Polymeric NP-Based Vaccines

Polymeric NPs, like lipid NPs, are widely used in gene therapy [160]. In terms of origin, polymeric NPs are divided into natural and synthetic polymers, both of which have shown remarkable potential for delivering RNA and DNA vaccines orally, overcoming the harsh conditions [161]. Most synthetic polymers used, such as PLGA are FDA-approved for drug and antigen delivery due to their biocompatibility, biodegradability, and ability to protect nucleic acids from enzymatic degradation [162]. PLGA-encapsulated DNA vaccines encoding *M. tuberculosis* in combination with Monophosphoryl lipid A (MPL) demonstrated the stimulation of a Th17 immune response, supporting PLGA as a potential delivery vehicle for vaccines against pathogens [163]. Similarly, oral PLGA NPs vaccines delivered Ovalbumin (OVA) that conjugated ulex europaeus agglutinin-1 (UEA-1) and encapsulated adjuvants like monophosphoryl lipid A (MPLA). Oral vaccination of mice with UEA-MPL/PLGA lipid complex protected OVA in the GI tract from exposure and induced both humoral and mucosal immunity [164].

Natural polymers such as chitosan and alginate are also important vehicles for RNA and DNA vaccine delivery. Chitosan, known for its mucoadhesive properties, enhances antigen uptake by mucosal tissues and opens epithelial tight junctions for improved translocation [165]. Chitosan NPs conjugated with M-cell targeting peptides, such as CVE30, and encapsulating DNA vaccines encoding *Coxsackievirus B3* antigens increased mucosal IgA production and T-cell activation upon oral administration in mice [166]. Xu et al. demonstrated increased systemic IgG and mucosal IgA responses in rats immunized with GI-stable chitosan nanoparticles modified with mannosylated and Eudragit^®^ and encapsulating bovine serum albumin [167]. Additionally, orally administrated alginate–chitosan-coated calcium phosphate NPs encapsulating OVA showed stability in acidic environments, efficiently targeting Peyer’s patches and eliciting systemic and mucosal responses [168]. These findings highlight the potential of polymer-based NPs to provide controlled release, enhance stability, and induce robust immune responses, positioning them as key players in advancing oral RNA and DNA vaccine technologies (Table 2).

### 7.4. Inorganic NP-Based Vaccines

Inorganic NPs such as gold (Au), silver (Ag), and calcium phosphate have demonstrated potential as carriers and adjuvants for RNA and DNA vaccines, owing to their rigid structures, ease of fabrication, and ability to enhance immune responses [175]. Chitosan–gold NPs (CsAuNPs) combined with extract (QS) as an adjuvant showed a remarkable 28-fold increase in tetanus toxoid (TT)-specific IgG and IgA responses in mice compared to controls [169]. Similarly, the use of *Asparagus racemosus* extract (ARE) as an adjuvant with CsAuNPs in oral TT delivery significantly boosted both systemic and mucosal immune responses without altering the physicochemical or antigenic properties of the NPs or TT [170]. Silver NPs (AgNPs), synthesized through green methods enhance the immunogenicity of antigens [171]. Further, AgNPs synthesized with *Eucalyptus* leaf extract enhanced the immune response to an inactivated rabies virus vaccine in a murine model, demonstrating adjuvanticity comparable to commercial alum [176]. Moreover, AgNPs, synthesized with PEG and β-D-glucose, encapsulating the H5 DNA vaccine significantly increased immune response to H5 and induced cytokine production in chicks after 1 h of immunization and did not produce toxicity (Table 2) [121].

### 7.5. ISCOMs

Immune-stimulatory complexes (ISCOMs) are 30–40 nm in size and contain cholesterol, phospholipids, Quil A (a saponin), and antigens, forming a strong adjuvant delivery system [177]. Their unique structure allows the incorporation of hydrophilic RNA or DNA antigens, facilitating robust immune responses. ISCOM-based vaccines have demonstrated efficacy across multiple routes of administration, including oral, subcutaneous, and intranasal, with the capacity to induce both humoral and cellular immune responses. For instance, oral administration of ISCOMs encapsulated influenza *A/Sichuan/87* antigens induced systemic IgG2a and local IgA antibodies, offering protection against homologous viral challenges [172]. Similarly, ISCOMs containing *Herpes simplex virus type 2* (HSV-2) antigens induced strong systemic and mucosal responses in mice, conferring protection against lethal viral doses [173]. ISCOM formulations incorporating DNA-based antigens, such as the OVA peptide epitope linked to a mucosal adjuvant (CTA1-DD), triggered robust T-cell responses alongside IgG2a and IgG1 isotype production, demonstrating their potential for inducing both cellular and humoral immunity [174]. Additionally, ISCOMATRIX™, a variation of ISCOMs, allows antigens to be added at later stages, providing flexibility in vaccine formulation (Table 2) [178].

## 8. Targeted Immunotherapies Using Orally Delivered DNA/RNA Nanoparticles

DNA/RNA NPs have shown promise in immunomodulation and oral delivery. It offers novel solutions for conditions such as allergies, autoimmune diseases, and chronic inflammatory disorders.

### 8.1. Food Allergies

Food allergies are an increasing global health issue, impacting millions of people and often causing serious allergic reactions. Current treatments mainly focus on avoiding allergens and managing symptoms, but they do not address the root cause of the immune system’s overreaction. DNA/RNA NPs are showing great promise as innovative tools for immunotherapy. These NPs can potentially reshape immune responses and help develop tolerance to allergens through non-invasive oral delivery systems [179].

Chitosan NPs have been leading the way in innovative allergy treatments. For example, studies have demonstrated the efficacy of chitosan NPs carrying allergen genes, such as the peanut allergen gene (pCMVArah2), have been effective in murine models of peanut allergy [180]. These NPs were able to significantly reduce allergic reactions, including lower IgE levels, decreased histamine release, and less vascular leakage. Significantly, they shifted the immune response from a Th2-dominated (allergic) state to a Th1-dominated (tolerant) profile, which is critical for long-term immune tolerance to allergens [181]. The choice of chitosan as a delivery material is based on its mucoadhesive properties, which prolong nanoparticle retention in the gut and promote uptake by Peyer’s patches. Moreover, chitosan can transiently open epithelial TJs and has been shown to promote tolerogenic DC activation and Treg differentiation, supporting its application in oral allergen-specific immunotherapy [182,183,184].

Similarly, RNA-based nanoparticles using materials such as chitosan, PLGA, or lipid carriers have been explored for food allergy immunotherapy by inhibiting the expression of cytokines involved in allergic inflammation. For instance, siRNA-loaded NPs targeting IL-4 and IL-13 have been shown to suppress pathways associated with IgE production and mast cell activation. In preclinical models, these RNA NPs reduced inflammatory markers and reduced allergic reactions, demonstrating their potential for food allergy treatment [33]. PLGA enables sustained release of siRNA, supporting prolonged immunomodulatory effects, while lipid NPs enhance cellular uptake and protect fragile RNA molecules from degradation. These design choices reflect a thoughtful approach to overcoming intestinal barriers and effectively modulating allergic responses at mucosal sites [185,186].

### 8.2. Autoimmune Diseases

Autoimmune diseases are chronic conditions where the immune system mistakenly attacks the body’s own tissues, causing inflammation, organ damage, and loss of function [187]. Current treatments, such as immunosuppressive drugs, focus on controlling symptoms and slowing disease progression but often come with significant side effects and do not address the underlying immune dysregulation [188]. DNA/RNA NPs have emerged as innovative tools in autoimmune disease therapy, offering the potential to modulate immune responses and restore immune tolerance through oral delivery systems.

Chitosan NPs have shown promise in targeting autoimmune diseases like Type 1 diabetes. For instance, oral delivery of DNA NPs encoding proinsulin has demonstrated significant efficacy in preclinical models. These NPs target GALT, inducing the generation of Tregs that suppress autoreactive T cells attacking pancreatic β-cells. Treated mice exhibited preserved β-cell function and delayed disease progression, underscoring the potential of this approach for immune tolerance induction in Type 1 diabetes [189].

Similarly, cationic RNA NPs have been explored for systemic lupus erythematosus. These NPs are designed to scavenge extracellular DNA and RNA, critical in forming immune complexes that drive inflammation in systemic lupus erythematosus. Their positive surface charge allows for electrostatic binding of negatively charged nucleic acids, thereby reducing immune activation and mitigating systemic inflammation [190].

RNA lipid NPs delivering siRNA or antisense oligonucleotides have also been investigated for rheumatoid arthritis. These NPs silence genes encoding for pro-inflammatory cytokines like IL-6 and TNF-α, key drivers of inflammation and joint destruction in rheumatoid arthritis. Lipid NPs are advantageous in this context due to their high transfection efficiency, biocompatibility, and ability to protect siRNA from enzymatic degradation. Preclinical studies have demonstrated reduced cytokine levels and joint inflammation alleviation, highlighting RNA NPs’ potential to modulate immune responses and protect against tissue damage [33].

### 8.3. Chronic Inflammatory Diseases

Chronic inflammatory diseases like inflammatory bowel disease and colitis involve long-lasting inflammation that can damage tissues and disrupt organ function. Treatments like corticosteroids and immunosuppressants are commonly used to manage symptoms and reduce inflammation, but they often fall short of addressing the underlying issues with the immune system. DNA/RNA NPs have shown promise as innovative therapies for these conditions. Delivered orally, these nanoparticles can target and modulate specific inflammatory pathways, offering a more precise and potentially effective way to treat these diseases [191,192].

Oral siRNA-loaded chitosan NPs targeting NF-κB, a central regulator of inflammation, have demonstrated significant efficacy in preclinical models of inflammatory bowel disease. These NPs helped lower levels of pro-inflammatory cytokines like TNF-α and IL-6, easing intestinal inflammation and promoting the repair of the epithelial lining [193]. Moreover, recent studies have explored using mucus-penetrating NPs to treat gastrointestinal inflammation conditions such as colitis. Silk fibroin-based NPs functionalized with Pluronic F127 have been developed to enhance mucus penetration. Pluronic modification enhances mucus penetration and cellular uptake, while the biodegradable silk fibroin core allows for sustained release of anti-inflammatory agents. These NPs were designed to deliver therapeutic agents directly to epithelial and immune cells, suppressing pro-inflammatory cytokines such as TNF-α. In preclinical models, this approach significantly reduced inflammation and improved mucosal healing, highlighting the potential of RNA-based therapies for managing colitis [194].

## 9. Future Outlook

As was described above, there is ample historical evidence that supports the oral route to modulate immune responses. A repeated ingestion of low doses of allergen leads to immune tolerance, which is widely used to treat food intolerance [195]. The low doses required to induce immune tolerance are suitable for the oral administration of RNA or DNA NPs since current nucleic acid NPs do not induce a high level of transgene expression [196]. Therefore, this strategy may have potential applications in reducing local inflammation, such as in Crohn’s disease or inflammatory bowel disease [197]. Further, the expression of antigens in the gut epithelium may induce immune tolerance to antigens and thus have potential applications to treat autoimmune diseases, such as diabetes [139,198]. Of relevance, repeated subcutaneous injections of an insulin peptide delayed the onset of diabetes in prediabetic individuals for at least 1 year [199]. Understanding the context of antigen presentation is critical for favorably modulating the immune response [200]. It is anticipated that the antigen may have to be designed to suit the individual MHC haplotype of the patient [201]. Perhaps new technologies such as synthetic biology, sequencing, and artificial intelligence may assist in designing effective and personalized treatments [202]. Given the importance of the immunology context, the co-expression of antigens together with anti- or pro-inflammatory molecules may achieve an even more effective modulation of the immune response.

Ingested antigens have successfully been used to modulate immune responses; however, they are degraded and broken down before intestinal absorption [11]. The expression of proteins encoded by RNA or DNA in gut epithelial cells might present intact antigens to intestinal immune cells, which is a potential advantage [203].

In summary, many routes of administration have been used to induce immune responses. Oral administration is minimally invasive and may lead to better adherence by patients, which is an important consideration when repeated administrations are required [204]. In contrast, other routes, such as intramuscular, are more immunogenic. In contrast, oral administration provides more opportunities to induce immune tolerance, perhaps even potentially inhibiting unwanted immune responses [205].

## Figures and Tables

**Figure 1 pharmaceutics-17-00609-f001:**
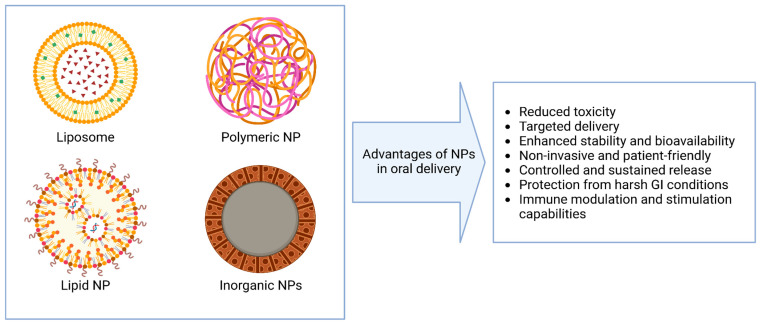
Advantages of various types of nanoparticles in the context of oral delivery.

**Figure 2 pharmaceutics-17-00609-f002:**
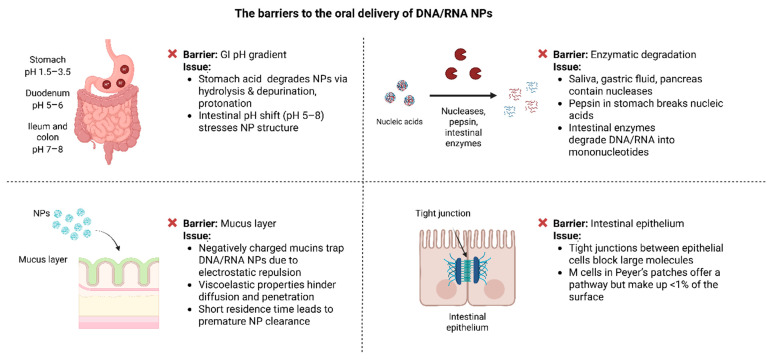
Biological barriers affecting the oral delivery of DNA/RNA nanoparticles.

**Table 1 pharmaceutics-17-00609-t001:** Strengths and limitations of various nanoparticle platforms for the oral delivery of nucleic acids.

Nanocarrier Type	Strengths	Limitations
Liposomes	Biodegradable and biocompatible [48] Encapsulate DNA/RNA efficiently [49] Protect nucleic acids from enzymatic degradation [49] Promote mucosal and systemic immunity [50]	Low stability in GI tract [51] Susceptible to oxidation/hydrolysis [52] Rapid clearance [53]
Lipid NPs (LNPs, SLNs, NLCs)	Effective for mRNA and saRNA delivery [54] Enable lymphatic absorption [55,56] Scalable and clinically validated (e.g., COVID-19 vaccines) [57,58,59]	Limited cargo flexibility [60] Potential lipid toxicity [61,62] Instability of some formulations [63,64]
Inorganic NPs (e.g., AuNPs, AgNPs, CaP)	Stable and robust structure [65] Functionalizable with nucleic acids [66,67] Strong adjuvant effect [68,69,70]	Poor biodegradability [71] Long-term accumulation risk [72,73] Potential cytotoxicity [74,75]
Polymeric NPs (e.g., PLGA, chitosan, alginate)	Protect DNA/RNA from acidic and enzymatic degradation [76,77] Mucoadhesive and target GALT (e.g., chitosan) [21,78,79] Controlled release [80,81,82,83] Biocompatible and biodegradable [84,85,86]	Complex synthesis [87,88] Batch variability [89,90] Low transfection efficiency [9,91]

**Table 2 pharmaceutics-17-00609-t002:** Summary of RNA/DNA nanoparticle platforms and applications for oral delivery vaccines.

Nanocarrier Type	Gene/Antigen Encoded	Target Disease	Experimental Model	References
Liposomes	Ag85A	Mycobacterium tuberculosis	Mice	[91]
Liposomes	M1 gene	Influenza A	Mice	[149]
Liposomes	SefA protein	Salmonella Enteritidis infection, Newcastle Disease	Chickens	[150]
Lipid NPs	SARS-CoV-2 spike protein	COVID-19	Humans	[58,157]
Lipid NPs	SSARS-CoV-2 spike protein	COVID-19 (Alpha, Delta)	Mice	[158]
Lipid NPs	SSARS-CoV-2 spike protein	COVID-19 (Alpha, Delta)	Mice	[159]
PLGA NPs	MPT83 protein	Mycobacterium tuberculosis	Mice	[163]
PLGA NPs	OVA	Not applicable	Mice	[164]
Chitosan NPs	Coxsackievirus B3 antigen	Viral myocarditis	Mice	[166]
Chitosan NPs	BSA	Not applicable	Rats	[167]
CaP NPs	OVA	Not applicable	Mice	[168]
Chitosan-functionalized gold NPs	Tetanus toxoid	Tetanus	Mice	[169,170]
Silver NPs	Inactivated rabies virus	Rabies	Mice	[171]
Silver NPs	H5 antigen	Avian influenza	Chicks	[121]
ISCOMs	Influenza A/Sichuan/2/87 surface glycoprotein subunit antigens	Influenza A	Mice	[172]
ISCOMs	HSV-2 subunit antigens	Herpes Simplex Virus Type 2 (HSV-2)	Mice	[173]
ISCOMs	OVA + CTA1-DD	Not applicable	Mice	[174]

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
