# Peer review of "Immune Modulation with Oral DNA/RNA Nanoparticles"

_pharmaceutics, 2025, doi:10.3390/pharmaceutics17050609_

Round 1
Reviewer 1 Report
Comments and Suggestions for Authors
The authors introduced the application of different carrier types of oral DNA/RNA vaccines in two directions: immune stimulation and immune tolerance. They first briefly discussed the physiological barriers that oral vaccines encounter during delivery. Then, they described the use of DNA and RNA-based nanoparticles (NPs), liposome-based NPs, lipid nanoparticles, polymeric-based NPs, inorganic-based NPs, and immune-stimulatory complex NPs in immune stimulatory oral vaccination. However, during this discussion, the authors did not clearly define these different delivery carriers or explain their characteristics, advantages, and disadvantages. For example, in paragraphs 118-126, the authors did not clearly define what DNA and RNA-based nanoparticles are.
Finally, the authors discussed the application of oral vaccines that can induce antigen-specific immune tolerance in food allergy immunotherapy, autoimmune disease therapy, and chronic inflammatory disease therapy. In these sections, the authors only introduced oral vaccines that achieve antigen-specific immune tolerance by altering the target of nucleic acids but did not elaborate on the design principles of delivery carriers in antigen-specific immune tolerance vaccines.
Author Response
Comments 1: The authors introduced the application of different carrier types of oral DNA/RNA vaccines in two directions: immune stimulation and immune tolerance. They first briefly discussed the physiological barriers that oral vaccines encounter during delivery. Then, they described the use of DNA and RNA-based nanoparticles (NPs), liposome-based NPs, lipid nanoparticles, polymeric-based NPs, inorganic-based NPs, and immune-stimulatory complex NPs in immune stimulatory oral vaccination. However, during this discussion, the authors did not clearly define these different delivery carriers or explain their characteristics, advantages, and disadvantages. For example, in paragraphs 118-126, the authors did not clearly define what DNA and RNA-based nanoparticles are.
Response 1: We appreciate the reviewer’s insightful feedback. In response, we have substantially revised the section discussing nanoparticle delivery systems (Section 3. Nanoparticle systems for oral nucleic acid delivery). Specifically, we have added a clear and concise definition of DNA/RNA-based nanoparticles (Lines 107-110).
In addition, we have included a new comparative Table 1, which summarizes the key properties, strengths, and limitations of the major nanocarrier types used in oral DNA/RNA delivery. This table highlights parameters such as biodegradability, mucosal penetration, enzymatic stability, toxicity, and immune activation potential, each supported by up-to-date references from peer-reviewed literature (Table 1). Furthermore, Figure 1 has been added to provide a visual overview. In the revised text, we also more clearly describe the structural and functional distinctions between each carrier type, and their specific suitability for oral nucleic acid delivery in the context of overcoming gastrointestinal barriers and promoting either immune activation or immune tolerance. We believe these revisions provide the necessary clarity and depth to address the reviewer’s concerns.
Comments 2: Finally, the authors discussed the application of oral vaccines that can induce antigen-specific immune tolerance in food allergy immunotherapy, autoimmune disease therapy, and chronic inflammatory disease therapy. In these sections, the authors only introduced oral vaccines that achieve antigen-specific immune tolerance by altering the target of nucleic acids but did not elaborate on the design principles of delivery carriers in antigen-specific immune tolerance vaccines.
Response 2: We thank the reviewer for this valuable observation. In response, we have revised Section 8 (Targeted immunotherapies using orally delivered DNA/RNA nanoparticles) of the manuscript to provide a clearer and more detailed explanation of the design principles of nanoparticle delivery systems used in antigen-specific immune tolerance vaccines. Specifically, for each disease category we have now included concise explanations of the rationale behind the choice of materials and nanoparticle design features.
In the updated text:
Lines 408-412: "The choice of chitosan as a delivery material is based on its mucoadhesive properties, which prolong nanoparticle retention in the gut and promote uptake by Peyer’s patches. Moreover, chitosan can transiently open epithelial TJs and has been shown to promote tolerogenic DC activation and Treg differentiation, supporting its application in oral allergen-specific immunotherapy [182], [183], [184]."
Lines 419-423: "PLGA enables sustained release of siRNA, supporting prolonged immunomodulatory effects, while lipid NPs enhance cellular uptake and protect fragile RNA molecules from degradation. These design choices reflect a thoughtful approach to overcoming intestinal barriers and effectively modulating allergic responses at mucosal sites [185], [186]."
Lines 470-475: "Pluronic modification enhances mucus penetration and cellular uptake, while the biodegradable silk fibroin core allows for sustained release of anti-inflammatory agents. These NPs were designed to deliver therapeutic agents directly to epithelial and immune cells, suppressing pro-inflammatory cytokines such as TNF-α. In preclinical models, this approach significantly reduced inflammation and improved mucosal healing, highlighting the potential of RNA-based therapies for managing colitis [194]."
We believe these revisions address the reviewer’s concern and improve the scientific depth of the discussion.
Reviewer 2 Report
Comments and Suggestions for AuthorsТhe authors present a short review focused on the oral delivery of nucleic acid containing nanoparticles for potential immunotherapeutic applications. Initially, the physiological barriers of nanoparticles for oral nucleic acid drug delivery are very briefly outlined, followed by an overview of various types of DNA/RNA nanocarriers for oral application and exhibiting potential immunomodulatory effect. Finally, the advantages and perspectives of nanoparticle-based nucleic acid oral administration for immune modulation are discussed. The proposed manuscript could be of interest to the readers of Pharmaceutics. However, I have serious concerns, mainly regarding the length and the lack of more in-depth discussion in the presented paper.
- First of all, I think that the manuscript is too short for a review paper. I understand that now there are no limits for minimum words and number of figures for a review or research article submitted to MDPI journals. However, a review paper with approx. 3000 words in total and just one figure, in my opinion, is not acceptable. The discussion in all sections could be easily extended and more figures could be added in order to make the review more informative and useful to the readers.
- I would recommend authors to pay attention to the title of section 3 and the corresponding subsections. The title of section 3 is: “Mechanisms by Which NPs Induce Immune Tolerance Versus Activation”. There are a couple of sentences about that. However, the following subsections are dedicated to the various types of nanoparticles for oral DNA/RNA administration (subsection 3.1) and their applications (subsection 3.2.). Therefore, I would recommend authors to reconsider the title of section 3 of the manuscript in order to be in line with the corresponding subsections.
- Page 5 (lines 149-153): A reference concerning the discussed work of Mohammadi et al. is missing. Moreover, I think that the cited reference 52 is not related to the discussed matter.
- In several occasions the cited literature does not illustrate oral administration of DNA/RNA containing nanoparticles which is the topic of the review. For example:
- page 5 (lines 146-148): references 50 and 51 concerning the Moderna and Pfizer-BioNTech’s COVID 19 vaccines;
- page 7, line 269: reference 82 explores antibody intravenous administration into mice.
Author Response
Comments 1: Тhe authors present a short review focused on the oral delivery of nucleic acid containing nanoparticles for potential immunotherapeutic applications. Initially, the physiological barriers of nanoparticles for oral nucleic acid drug delivery are very briefly outlined, followed by an overview of various types of DNA/RNA nanocarriers for oral application and exhibiting potential immunomodulatory effect. Finally, the advantages and perspectives of nanoparticle-based nucleic acid oral administration for immune modulation are discussed. The proposed manuscript could be of interest to the readers of Pharmaceutics. However, I have serious concerns, mainly regarding the length and the lack of more in-depth discussion in the presented paper.
First of all, I think that the manuscript is too short for a review paper. I understand that now there are no limits for minimum words and number of figures for a review or research article submitted to MDPI journals. However, a review paper with approx. 3000 words in total and just one figure, in my opinion, is not acceptable. The discussion in all sections could be easily extended and more figures could be added in order to make the review more informative and useful to the readers.
Response 1: We appreciate this suggestion. Our initial aim was to synthesize a mini-review, hence the abbreviated nature of the manuscript. The manuscript has now been substantially expanded and the updated draft exceeds 6,000 words. Following the reviewer’s recommendation, we have added new visual materials: The previous Figure 1 has been replaced with a new Figure, which presents the content more clearly in higher resolution. An additional figure and two new tables have been added to enhance overall readability.
Comments 2: I would recommend authors to pay attention to the title of section 3 and the corresponding subsections. The title of section 3 is: “Mechanisms by Which NPs Induce Immune Tolerance Versus Activation”. There are a couple of sentences about that. However, the following subsections are dedicated to the various types of nanoparticles for oral DNA/RNA administration (subsection 3.1) and their applications (subsection 3.2.). Therefore, I would recommend authors to reconsider the title of section 3 of the manuscript in order to be in line with the corresponding subsections.
Response 2: We have revised the section title to "Design and applications of nanoparticles for inducing immune tolerance and activation" to better reflect the content of the subsections. (Now section 6, page 10, line 242)
Comments 3: Page 5 (lines 149-153): A reference concerning the discussed work of Mohammadi et al. is missing. Moreover, I think that the cited reference 52 is not related to the discussed matter.
Response 3: We apologize for this important oversight. The incorrect reference (formerly cited as reference 52) has now been replaced with the correct citation of the work by Mohammadi et al. on oral self-amplifying RNA Lipid NP vaccines, which is now listed as reference 158 (Now page 12, line 319).
Comments 4: In several occasions the cited literature does not illustrate oral administration of DNA/RNA containing nanoparticles which is the topic of the review. For example:
page 5 (lines 146-148): references 50 and 51 concerning the Moderna and Pfizer-BioNTech’s COVID 19 vaccines;
page 7, line 269: reference 82 explores antibody intravenous administration into mice.
Response 4: We thank the reviewer for this accurate remark. References 50 and 51 were originally included to inform readers that Lipid NP-based vaccines were initially approved for non-oral administration before their adaptation for oral use (Now page 12, line 318, references 58 and 157). A clarification has been included in the revised manuscript. Additionally, reference 82, which discussed intravenous antibody delivery, has been replaced with more relevant literature on oral delivery, now cited as references 139 and 198 (Now page 18, line 486).
Reviewer 3 Report
Comments and Suggestions for Authors
The review article "Immune Modulation with Oral DNA/RNA Nanoparticles" by Kart et al. offers a promising overview of a relevant and rapidly developing field. The authors address key aspects of oral DNA/RNA nanoparticle delivery for immune modulation, including mechanisms, design, delivery strategies, and potential therapeutic applications. However, several substantial revisions are required before the article is suitable for publication.
Firstly, the scope of the literature review needs significant expansion. While 88 references are cited, comprehensive reviews typically include upwards of 150-200 citations to thoroughly capture the breadth of existing research. A more exhaustive literature search is crucial to ensure a complete and nuanced representation of the current knowledge.
Secondly, the visual presentation requires considerable improvement. Figure 1 lacks clarity and informativeness and needs to be redesigned for better visual impact and conveyance of information. More importantly, the review would greatly benefit from the addition of several schematic figures and diagrams. Visual summaries of key concepts, such as nanoparticle design principles, mechanisms of immune modulation, and different delivery strategies, would significantly enhance reader comprehension and provide a more accessible overview of this complex topic. Graphical representations of data, where appropriate, would also strengthen the presentation.
Finally, while the authors touch upon the challenges of stability, delivery efficiency, and safety, a more in-depth analysis of these limitations is needed. This should include specific examples of these challenges encountered in research and potential strategies to overcome them. A more detailed comparison of different nanoparticle platforms and their suitability for oral delivery would also be valuable (in Table form).
In summary, while the review presents a promising foundation, it requires significant revisions, including expanding the literature review, improving the visual presentation with additional Figures and diagrams, Tables, and providing a more critical analysis of the challenges and opportunities in the field. Therefore, I recommend accepting this article for publication only after these substantial revisions are addressed.
Author Response
Comments 1: The review article "Immune Modulation with Oral DNA/RNA Nanoparticles" by Kart et al. offers a promising overview of a relevant and rapidly developing field. The authors address key aspects of oral DNA/RNA nanoparticle delivery for immune modulation, including mechanisms, design, delivery strategies, and potential therapeutic applications. However, several substantial revisions are required before the article is suitable for publication.
Firstly, the scope of the literature review needs significant expansion. While 88 references are cited, comprehensive reviews typically include upwards of 150-200 citations to thoroughly capture the breadth of existing research. A more exhaustive literature search is crucial to ensure a complete and nuanced representation of the current knowledge.
Response 1: We thank the reviewer for this valuable suggestion. In response, we have expanded the reference list from 88 to 205 citations, incorporating a broader range of recent and foundational studies across all sections of the manuscript. Key updates were made in the sections on nanocarrier types, mucosal immunity, and disease-specific applications. This revision ensures a more comprehensive and balanced overview of the current state of research in oral DNA/RNA nanoparticle-based immune modulation.
Comments 2: Secondly, the visual presentation requires considerable improvement. Figure 1 lacks clarity and informativeness and needs to be redesigned for better visual impact and conveyance of information. More importantly, the review would greatly benefit from the addition of several schematic figures and diagrams. Visual summaries of key concepts, such as nanoparticle design principles, mechanisms of immune modulation, and different delivery strategies, would significantly enhance reader comprehension and provide a more accessible overview of this complex topic. Graphical representations of data, where appropriate, would also strengthen the presentation.
Response 2: We thank the reviewer for this insightful suggestion. In response, we have redesigned and clarified the former Figure 1, which now appears as Figure 3 in the revised manuscript. This updated figure provides a more structured schematic comparing the structural and functional features of different nanoparticle types used in oral DNA/RNA delivery.
Additionally, we have added the following new schematic figures to enhance visual clarity and reader comprehension:
- Figure 1. Advantages of various types of nanoparticles in the context of oral delivery – a concise visual summary of key properties (e.g., mucoadhesion, enzymatic protection, targeting ability).
- Figure 2. Biological barriers affecting oral delivery of DNA/RNA nanoparticles – an overview of the physiological and immunological challenges encountered during oral administration.
We have also expanded the use of tables to present comparative data and key findings in a reader-friendly format:
- Table 1. Strengths and limitations of various nanoparticle platforms for oral delivery of nucleic acids – a comparative summary based on criteria such as biodegradability, toxicity, mucosal immunity, and stability.
- Table 2. Summary of RNA/DNA nanoparticle platforms and applications for oral delivery vaccines – highlighting carrier types, disease targets, delivery strategies, and immunological outcomes.
We believe these visual and structural enhancements significantly improve the clarity, accessibility, and overall presentation of the review.
Comments 3: Finally, while the authors touch upon the challenges of stability, delivery efficiency, and safety, a more in-depth analysis of these limitations is needed. This should include specific examples of these challenges encountered in research and potential strategies to overcome them. A more detailed comparison of different nanoparticle platforms and their suitability for oral delivery would also be valuable (in Table form).
In summary, while the review presents a promising foundation, it requires significant revisions, including expanding the literature review, improving the visual presentation with additional Figures and diagrams, Tables, and providing a more critical analysis of the challenges and opportunities in the field. Therefore, I recommend accepting this article for publication only after these substantial revisions are addressed.
Response 3: We thank the reviewer for this comprehensive and constructive feedback. In response, we have addressed the limitations of oral DNA/RNA nanoparticle delivery more thoroughly throughout the manuscript.
Specifically:
- In Sections 3 and 4, we now discuss in greater detail the common challenges associated with stability in the gastrointestinal tract, low delivery efficiency, and potential toxicity, with specific examples from recent studies.
- We have also included descriptions of emerging strategies to overcome these limitations, such as surface modifications, mucoadhesive coatings, targeting ligands, and pH-responsive systems.
To provide a clearer comparison of nanoparticle systems:
- We have added Table 1. Strengths and limitations of various nanoparticle platforms for oral delivery of nucleic acids, which presents a side-by-side evaluation of liposomes, Lipid NPs, polymeric NPs, and inorganic NPs.
These revisions aim to deliver a more critical and balanced view of both the opportunities and barriers in the field. We believe this updated version addresses the reviewer’s concerns and significantly improves the depth and utility of the review.
Round 2
Reviewer 1 Report
Comments and Suggestions for Authors
The authors have improved the quality of the review and addressed my major concerns.
Author Response
Comments 1: The authors have improved the quality of the review and addressed my major concerns.
Response 1: We sincerely thank the reviewer for their positive feedback. We are glad to hear that the revisions have addressed the major concerns and have improved the quality of the review. Your constructive comments have been invaluable in refining the manuscript, and we appreciate the time and effort you have dedicated to reviewing our work.
Reviewer 2 Report
Comments and Suggestions for Authors
The authors have made significant efforts to improve their manuscript according to my remarks and suggestions. Therefore, in my opinion, the manuscript is now suitable for publication as a review paper in its current form.
Author Response
Comments 1: The authors have made significant efforts to improve their manuscript according to my remarks and suggestions. Therefore, in my opinion, the manuscript is now suitable for publication as a review paper in its current form.
Response 1: We sincerely thank the reviewer for the positive feedback and recognition of the efforts made in revising the manuscript. We greatly appreciate the constructive remarks and suggestions, which have significantly enhanced the quality of the paper. We are pleased to hear that the manuscript is now considered suitable for publication as a review paper in its current form. We remain grateful for the reviewer’s time and valuable input.
Reviewer 3 Report
Comments and Suggestions for Authors
The authors have taken my comments into consideration and made significant improvements to the manuscript. Nevertheless, there remains a need to address some stylistic inconsistencies and issues related to the formatting of subheadings and table captions. Additionally, it would be advisable to revise the abstract to emphasize the key points of the paper. Once these revisions have been made, the manuscript is ready for publication in the journal.
Author Response
Comments 1: The authors have taken my comments into consideration and made significant improvements to the manuscript. Nevertheless, there remains a need to address some stylistic inconsistencies and issues related to the formatting of subheadings and table captions.
Response 1: We appreciate the reviewer’s acknowledgment of the improvements made to the manuscript. In response to the remaining concerns:
- Stylistic inconsistencies: We carefully reviewed the entire manuscript for stylistic consistency. All terminological inconsistencies were addressed, particularly regarding the use of terms like "DNA/RNA nanoparticles". These terms have been standardized throughout the manuscript to avoid confusion. We also ensured that all abbreviations (e.g., PLGA, PEI, PCL) are expanded upon their first use and consistently used thereafter.
-
Formatting of subheadings: All subheadings have been revised to ensure consistent title case and proper spacing. We also ensured consistent formatting across main headings and subsections.
- Table captions: Table captions have been reviewed and corrected for consistency. They now follow standard formatting rules, with captions placed above the tables.
Comments 2: Additionally, it would be advisable to revise the abstract to emphasize the key points of the paper. Once these revisions have been made, the manuscript is ready for publication in the journal.
Response 2: We have revised the abstract to emphasize the key points of the paper, as per the reviewer’s suggestion. The updated abstract now provides a clearer and more concise summary of the main objectives. New abstract:
"Oral delivery of DNA/RNA nanoparticles represents a transformative approach in immunotherapy and vaccine development. These nanoparticles enable targeted immune modulation by delivering genetic material to specific cells in the gut-associated immune system, triggering both mucosal and systemic immune responses. Unlike parenteral administration, the oral route offers a unique immunological environment that supports both tolerance and activation, depending on formulation design. This review explores the underlying mechanisms of immune modulation by DNA/RNA nanoparticles, their design and delivery strategies, and recent advances in their application. Emphasis is placed on strategies to overcome physiological barriers such as acidic pH, enzymatic degradation, mucus entrapment, and epithelial tight junctions. Special attention is given to the role of gut-associated lymphoid tissue in mediating immune responses and the therapeutic potential of these systems in oral vaccine platforms, food allergies, autoimmune diseases, and chronic inflammation. Despite challenges, recent advances in nanoparticle formulation support the translation of these technologies into clinical applications for both therapeutic immunomodulation and vaccination."
We believe these revisions strengthen the abstract and offer a clearer overview of the paper's content.